

# A novel approach to recognition of Alzheimer's and Parkinson's diseases: random subspace ensemble classifier based on deep hybrid features with a super-resolution image

Adi Alhudhaif

Department of Computer Science, College of Computer Engineering and Sciences in Al-kharj, Prince Sattam Bin Abdulaziz University, Al-kharj, Saudi Arabia

## ABSTRACT

**Background**. Artificial intelligence technologies have great potential in classifying neurodegenerative diseases such as Alzheimer's and Parkinson's. These technologies can aid in early diagnosis, enhance classification accuracy, and improve patient access to appropriate treatments. For this purpose, we focused on AI-based auto-diagnosis of Alzheimer's disease, Parkinson's disease, and healthy MRI images.

**Methods**. In the current study, a deep hybrid network based on an ensemble classifier and convolutional neural network was designed. First, a very deep super-resolution neural network was adapted to improve the resolution of MRI images. Low and high-level features were extracted from the images processed with the hybrid deep convolutional neural network. Finally, these deep features are given as input to the k-nearest neighbor (KNN)-based random subspace ensemble classifier.

**Results**. A 3-class dataset containing publicly available MRI images was utilized to test the proposed architecture. In experimental works, the proposed model produced 99.11% accuracy, 98.75% sensitivity, 99.54% specificity, 98.65% precision, and 98.70% F1-score performance values. The results indicate that our AI system has the potential to provide valuable diagnostic assistance in clinical settings.

## INTRODUCTION

Detecting Alzheimer's and Parkinson's diseases early can lead to better treatment and management of these neurological conditions. Artificial intelligence (AI) can play a crucial role in detecting these diseases by analyzing large amounts of data from various sources, including medical records, brain scans, genetic information, and other relevant data. Moreover, AI can help researchers develop new treatments and therapies for these diseases. By analyzing data from clinical trials and patient records, AI can identify potential drug candidates, predict patient outcomes, and improve the efficacy of clinical trials (*Htike et al. (2019)*; *Schork, 2019*; *Haleem, Javaid & Khan, 2019*; *Noor et al., 2020*).

Corresponding author
Adi Alhudhaif,
a.alhudhaif@psau.edu.sa

**Table 1  Previous studies based on Parkinson's disease (PD) and Alzheimer's disease (AD).**

| References | Konu | Dataset | Model |
|---|---|---|---|
| *Helaly, Badawy & Haikal (2022)* | Dementia of AD | 4 classes and 48,000 images | Fine-tuned VGG19 model |
| *Bhagat et al. (2023)* | Dementia of AD | 5 classes and 1,101 images | Fine-tuned MobileNet model |
| *Guan et al. (2019)* | Dementia of AD | 3 classes and 3,415 images | 3D 16-layer ResNet |
| *Murugan et al. (2021)* | Dementia of AD | 4 classes and 6,400 images | DEMNET architecture based on CNN |
| *Shahwar et al. (2022)* | AD and healthy control | 2 classes and 6,400 images | Hybrid CNN based on ResNet34 |
| *Sivaranjini & Sujatha (2020)* | PD and healthy control | 2 classes and 7,646 images | Fine-tuned AlexNet model |
| *Peng et al. (2017)* | PD and healthy control | 2 classes and 172 images | Multilevel ROI based features and multi-kernel SVM |
| *Raees & Thomas (2021)* | Mild Cognitive Impairment (MCI), AD, and Normal classes | 3 classes and 800 images | Vgg19 + svm |
| *Basnin et al. (2021)* | PD and healthy control | 2 classes and 1,387 images | DenseNet-LSTM |
| *AlSaeed & Omar (2022)* | AD and healthy control | 2 classes and 741 images | CNN-based feature extraction |
| *Savaş (2022)* | Normal, MCI, and AD | 3 classes and 4,306 images | Pre-trained EfficientNet |
| *Noella & Priyadarshini (2023a)* | AD, Frontotemporal Dementia (FTD), PD, and healthy control | 4 classes and 1,210 images | Generative adversarial deep convolutional neural network |
| *Noella & Priyadarshini (2023b)* | PD, AD, and healthy control | 3 classes and 1,050 images | Multi-feature extraction and classifier |
| *Alsharabi et al. (2023)* | PD, AD, and healthy control | 3 classes and 1,208 images | AlexNet–quantum learning |

The importance of using AI in detecting Alzheimer's and Parkinson's diseases lies in its ability to improve early detection, facilitate better treatment and management, and accelerate the development of new treatments and therapies. Additionally, early detection and treatment of Alzheimer's and Parkinson's can result in cost savings for healthcare systems. By detecting the disease early, patients may need fewer hospitalizations and interventions, leading to overall lower healthcare costs (*Joshi et al., 2010*; *Laske et al., 2015*; *Noor et al., 2020*).

Overall, using AI to detect Alzheimer's and Parkinson's diseases can revolutionize diagnosing and treating these conditions. Furthermore, by improving accuracy, facilitating early detection, and accelerating research, AI can potentially improve the lives of millions of people affected by these diseases. For this purpose, many studies have been carried out for the automatic detection of these diseases by using artificial intelligence techniques in the literature. These studies are summarized in Table 1.

Table 1 provides detailed information about previous studies for classifying AD and PD. In most of these studies, pre-trained convolutional neural networks based on transfer learning were utilized. In addition, in some studies, pre-trained architectures were used as feature extractors, and machine learning classifier methods such as support vector machine (SVM), and K-nearest neighbors (KNN) were employed as classifiers. In the remaining studies, features were obtained using machine learning feature extraction methods such as gray level co-occurrence matrice (GLCM), the histogram of oriented gradients (HOG), and local binary pattern (LBP), and classification was carried out with algorithms such as SVM, KNN, etc. On the other hand, in most of these studies, either Parkinson's or Alzheimer's diseases were addressed. In the studies of *Noella & Priyadarshini*

*(2023a)*; *Noella & Priyadarshini (2023b)*, and *Alsharabi et al. (2023)*, Parkinson's disease, Alzheimer's disease, and healthy control classes were considered as in the present study.

In this article, we proposed a KNN-based random subspace ensemble classifier model based on deep hybrid features with super resolution images to classify Alzheimer's and Parkinson's diseases. Firstly, a deep super-resolution neural network has been applied to increase the resolution of MRI images. Deep convolutional neural networks with different structures were adapted for these improved images, and low and high-level features were extracted. Finally, these features are given to the input of the developed KNN-based random subspace ensemble classifier model. The proposed model has demonstrated promising results in experimental studies to classify Alzheimer's and Parkinson's diseases.

**Listed below are the key contributions of the proposed model in the classification of Alzheimer's and Parkinson's diseases**

- The current study adapted deep super-resolution neural network architecture for MRI images. Thus, high-resolution, and enhanced MR images were obtained.
- A hybrid deep neural network extracted low and high-level features from MRI images.
- We presented a high-performance KNN-based random subspace ensemble classifier model in classifying AD, PD, and healthy control.
- Experimental studies have shown that our system is a more accurate, faster, less costly, and more objective approach. Based on these findings, it seems likely that our AI system could assist in diagnosing patients in clinical settings in the future.

The remainder of the article is organized as follows: the 'Method' section is dedicated to presenting the proposed methodology and theoretical background of the article, while the 'Material and Experimental results' section provides details on the dataset and experiment results. The 'Discussion' section compares the performance results of the proposed model with previous studies. Lastly, the 'Conclusions' section provides the final thoughts of the article.

# METHOD

In the present study, we proposed a deep neural network-based ensemble classifier approach for classifying Alzheimer's and Parkinson's diseases with MRI images. This approach comprises three phases: pre-processing, deep feature network, and classification. Figure 1 illustrates the overall representation of the proposed system, encompassing all the stages above. In addition, the stages that make up the proposed architecture are detailed in sub-tabs.

## Pre-processing

In the pre-processing phase, we focused on enhancing the images' quality and increasing their resolution. To achieve this goal, we utilized the very-deep super-resolution neural network, a deep learning model capable of generating high-resolution images from low-resolution images (*Kim, Lee & Lee, 2016*). This model leverages low-level and high-level features through a skip link to improve performance. It is specifically designed for

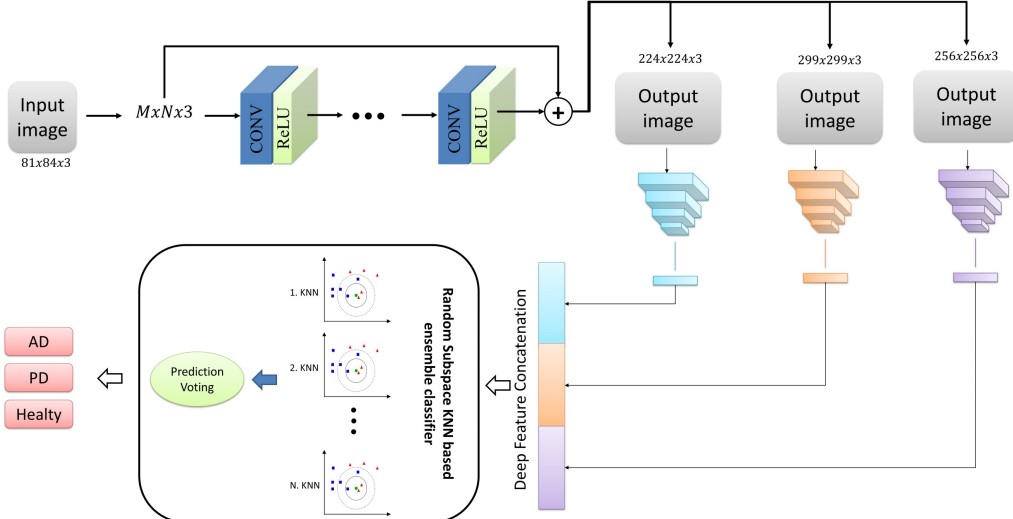

**Figure 1  The structure of the proposed architecture.**

image super-resolution tasks, which involve generating high-resolution images from low-resolution ones. The very-deep super-resolution neural network is a type of convolutional neural network (CNN) that can learn the mapping between low-resolution and high-resolution images (*Kim, Lee & Lee, 2016*; *Ooi & Ibrahim, 2021*).

The architecture of a very-deep super-resolution neural network typically consists of multiple layers of convolutional and up sampling layers. The network receives a low-resolution image as input and generates a high-resolution output image. During training, the network learns from paired examples of low-resolution and high-resolution images and strives to minimize the discrepancy between the predicted high-resolution image and the ground truth high-resolution image (*Dong, Loy & Tang, 2016*; *Peng et al., 2021*). The key feature of the very-deep super-resolution neural network is its depth. The network typically has many layers (sometimes over 100), allowing it to capture complex and high-level features in the input image. In addition, the network is often trained using residual learning, which means that the network is designed to learn the residual between the low-resolution and high-resolution images rather than the high-resolution image directly (*Kim, Lee & Lee, 2016*; *Ren et al., 2019*). This allows the network to focus on learning the high-frequency details lost in the low-resolution image. The very-deep super-resolution neural network has been shown to achieve state-of-the-art performance on various super-resolution tasks, including single-image super-resolution and video super-resolution. It has also been used in applications such as medical imaging, remote sensing, and surveillance (*Kim, Lee & Lee, 2016*; *Shi et al., 2016*; *Lim et al., 2017*; *Ooi & Ibrahim, 2021*).

In summary, the very-deep super-resolution neural network is a deep learning model capable of producing high-resolution images from low-resolution images. Its key feature is its depth, which allows it to capture complex and high-level features in the input image,

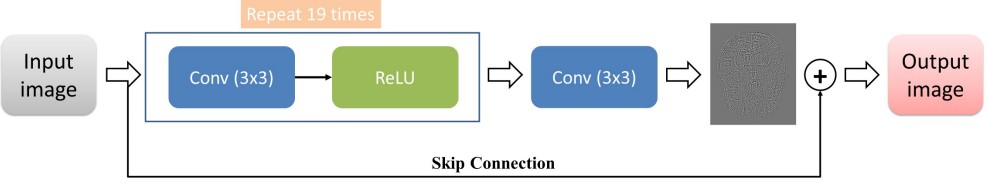

**Figure 2  The architecture of the VSDR model.**

and it is often trained using residual learning to focus on high-frequency details. The VSDR architecture was employed for the current study, which involves cascading convolutional layers with dimensions of $3 \times 3 \times 64$. Additionally, the image segment size was 41 by 41. Figure 2 presents the overall framework of this network architecture.

## Deep feature extraction

Multiple layers typically comprise a convolutional neural network, including convolutional, activation functions, pooling, and fully connected layers. At the beginning of a convolutional neural network is the convolutional layer, which utilizes a set of learnable filters to process the input image (*Kalchbrenner, Grefenstette & Blunsom, 2014*; *O'Shea & Nash, 2015*; *Albawi, TA & Al-Zawi, 2017*). Let x be the input image, and let w be a filter. The output of a convolutional layer is given by:

$$y(a,b,k) = \sum_m \sum_n \sum_c x(a+m, b+n, c) * w(m,n,c,k) \tag{1}$$

Here, $a$ and $b$ index the spatial location of the output feature map, $k$ indexes the filter, and $c$ indexes the input channels. The summation over $m$ and $n$ represents the sliding of the filter over the input image. The "$*$" represents the convolution operation.

The output of the convolutional layer is then passed through an activation function, such as the ReLU function (*Ramachandran, Zoph & Le, 2017*):

$$y(a,b,k) = \max(0, y(a,b,k)) \tag{2}$$

The output of the activation function is typically fed into a pooling layer, which reduces the dimensionality of the feature map by taking the maximum (or average) value within a local neighborhood (*Bayar & Stamm, 2016*). The pooling operation can be represented as follows:

$$y(a,b,k) = \max(x(a_{stride} + m, b_{stride} + n, k)) \tag{3}$$

Where stride is the distance between adjacent pooling regions, and $m$ and $n$ index the local neighborhood. The output of the pooling layer is then passed to the next convolutional layer, and the process is repeated until the final layer, which is typically a fully connected layer. Finally, the output of the last pooling layer is flattened into a vector, and the fully connected layer applies a set of learnable weights to the input vector to produce the final output scores:

$$y = Wx + b \tag{4}$$

**Table 2  The characteristics of deep networks used in this study.**

| Model | Fully connected layer name | Depth | Size | Parameters (millions) | Image input size |
|---|---|---|---|---|---|
| DarkNet53 | conv53 | 53 | 155 MB | 41.6 | 256 × 256 |
| DenseNet201 | fc1000 | 201 | 77 MB | 20.0 | 224 × 224 |
| Xception | prediction | 71 | 85 MB | 22.9 | 299 × 299 |

where $W$ is a matrix of weights and $b$ is a bias term. The output scores can then be passed through a softmax function to produce a probability distribution over the possible classes (*Kalchbrenner, Grefenstette & Blunsom, 2014*; *O'Shea & Nash, 2015*). Overall, convolutional neural networks are a powerful tool for image and video recognition tasks. Their ability to automatically learn feature representations from input data has led to significant advances in computer vision research.

The pre-trained architectures that include those mentioned above convolutional neural network layers offer many benefits, including faster training times, improved accuracy, transfer learning, and reduced data requirements. These benefits can make it easier to develop and deploy machine learning models for a wide range of tasks and applications (*Ağdaş, Türkoğlu & Gülseçen (2021)*; *Turkoglu, 2021*; *Uzen, Turkoglu & Hanbay, 2021*; *Imak et al., 2022*). Accordingly, we have adapted pre-trained convolutional neural networks based on transfer learning for Alzheimer's and Parkinson's disease detection problems. In the current study, we used three robust deep architectures with different structures, DarkNet53 (*Redmon & Farhadi, 2018*), DenseNet201 (*Huang et al., 2017*), and Xception (*Chollet, 2017*), and extracted low- and high-level features from MRI images using the learned weights of these architectures. Detailed information about these architectures is given in Table 2.

Table 2 was utilized to obtain pre-trained deep architectures that acted as feature extractors. Deep features were extracted using fully connected layers of deep architectures given in column 2 of Table 2. Then, these obtained features are combined and given to the input of the classifier model.

## Ensemble classification

In this study, we used a classification algorithm that combines the concepts of KNN and random subspace ensemble methods. KNN is a classification algorithm that predicts the class of a new instance based on the class labels of the K nearest instances in the training set. The algorithm assumes that similar instances have similar class labels (*Chen, Bi & Wang, 2006*; *Zhang & Zhou, 2007*; *Zhang et al., 2017*). On the other hand, random subspace ensemble methods are classification algorithms that create multiple subsets of the original feature set and train a classifier on each subset. The idea behind this approach is to reduce the variance of the classification model by introducing diversity among the classifiers (*Ho, 1998*; *Tremblay, Sabourin & Maupin, 2004*; *Demir, AM & Sengur, 2020*), *Önder, Dogan & Polat (2023)*

In KNN based random subspace ensemble classifier, KNN, and random subspace ensemble methods are combined to create a robust classification model. The algorithm

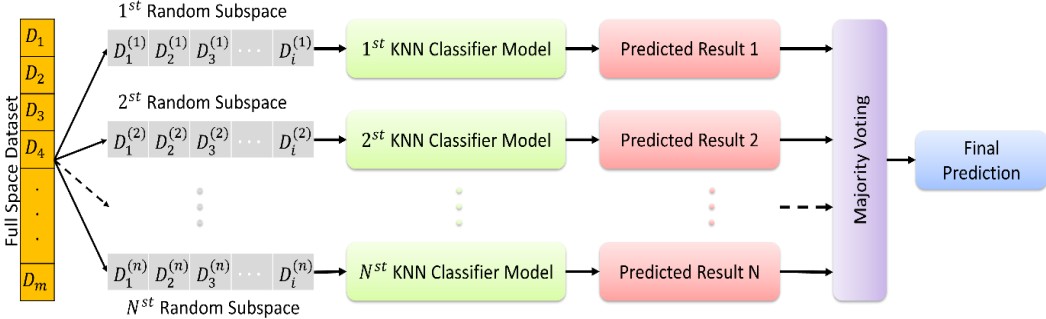

**Figure 3** **The random subspace ensemble classifier based on K-nearest neighbors.**

first creates multiple subsets of the original feature set using random subspace ensemble methods. Then, for each subset, the KNN algorithm is applied to obtain the class labels of the nearest neighbors. Finally, the class labels predicted by each KNN model are aggregated to obtain the final prediction. This proposed ensemble classifier is represented in Fig. 3.

The random subspace ensemble classifier based on K-nearest neighbors used in the present study is a machine learning algorithm that improves classification accuracy by combining a set of KNN classifiers, each trained on a randomly chosen subset of features, with $k = 1$. This algorithm aims to optimise the overall performance by combining multiple KNN classifiers, each trained on a randomly selected subset of features. This method aims to create a more powerful and generalisable classifier by increasing the diversity throughout the model, with each subset emphasising different attributes. In this way, it is possible to obtain more effective and reliable classification results in complex data sets.

## MATERIAL AND EXPERIMENTAL RESULTS

In this article, we proposed a KNN-based random subspace ensemble classifier model based on deep CNN to classify Alzheimer's and Parkinson's diseases. We conducted the experimental works using the MATLAB 2022b platform, utilizing a workstation computer equipped with an NVIDIA Quadro P4000 card and 32GB of RAM to perform the applications. To evaluate the models utilized in the experiment, the following metrics were used: accuracy, sensitivity, specificity, precision, and F1-score (see Eqs. (5)–(9)):

$$Accuracy = \frac{(TP + TN)}{(TP + TN + FP + FN)} \tag{5}$$

$$Sensitivity = \frac{TP}{(TP + FN)} \tag{6}$$

$$Specificity = \frac{(TN)}{(TN + FP)} \tag{7}$$

**Table 3   The training and testing set used in the current study.**

| | Training | | | Testing | | |
|---|---|---|---|---|---|---|
| | **AD** | **PD** | **Healthy** | **AD** | **PD** | **Healthy** |
| **Raw dataset** | 2,560 | 774 | 3,479 | 640 | 193 | 734 |
| **After augmentation** | 5,000 | 5,000 | 5,000 | 640 | 193 | 734 |

$$Precision = \frac{(TP)}{(TP + FP)} \tag{8}$$

$$F1 - Score = 2 * \frac{(Sensitivity * Precision)}{(Sensitivity + Precision)} \tag{9}$$

Here, true-negative (TN), true-positive (TP), false-negative (FN), and false-positive (FP) are denoted by their respective abbreviations.

## Material: Dataset

Our work involved utilizing a dataset comprising MRI images of individuals diagnosed with AD, PD, and Healthy controls. The dataset (https://www.kaggle.com/datasets/farjanakabirsamanta/alzheimer-diseases-3-class) was publicly available and a valuable resource in our study. On the other hand, the dataset was split into two subsets for our study, with 80% of the data used for training and 20% used for testing. Notably, our data augmentation methods were exclusively applied to the training set.

It has been determined that the number of images between classes in the dataset used in this study is not equal. Therefore, various data augmentation techniques, including random rotate (90), vertical flip, center crop, vertical translation ([-5 5]), contrast (1.2) and brightness (0.5) were employed to address the issue of overfitting, and stabilizing the dataset. Numerical information about the raw data and the data augmented by these methods are given in Table 3.

## RESULTS

In this article, we performed extensive experimental studies based on an ensemble classifier and deep neural network. In the first experimental work, we used a pre-trained deep model based on transfer learning approaches as fine-tuning and training from scratch. This experimental study investigated the effect of pre-trained weights on classification performance. Accordingly, the pre-trained DarkNet architecture based on fine-tuning and training from scratch approaches has been adapted for the classification of AD, PD, and Healthy Control. For this purpose, the first layers of this architecture were frozen and fully connected, and SoftMax and classification layers were added instead of the last three layers. In the fine-tuning process, pre-trained weights are used, while in another approach, training is performed from scratch. On the other hand, in the training phase of this deep architecture, deep parameters were set as epoch number 30, batch size 16, the

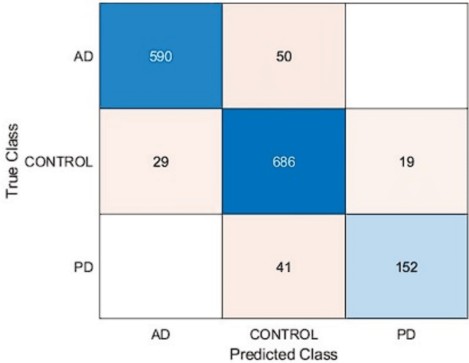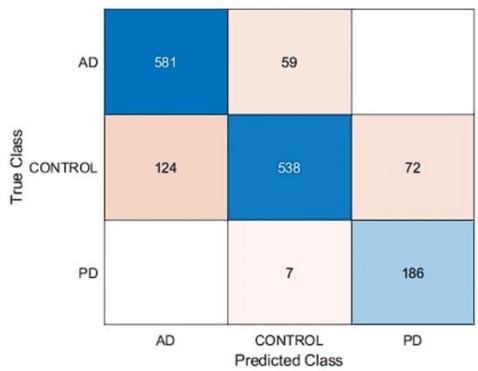

**Figure 4** **Confusion matrixes of the pre-trained model.** (A) Fine-tuning, (B) starch.

**Table 4** **Performance metrics (%) of pre-trained deep networks as a feature extractor.**

| Models | Accuracy | Sensitivity | Specificity | Precision | F1-score |
|---|---|---|---|---|---|
| DenseNet201 | 9,707 | 9,570 | 9,841 | 9,625 | 9,597 |
| DarkNet53 | 9,432 | 9,409 | 9,694 | 9,315 | 9,360 |
| Xception | 9,585 | 9,603 | 9,771 | 9,539 | 9,571 |

initial learning rate 0.0001, and optimization method Adam. Confusion matrices based on the results obtained using these approaches are given in Fig. 4.

According to the confusion matrices in Fig. 4, the DarkNet architecture based on the fine-tuning approach produced 91.13% accuracy, 94.85% specificity, and 88.13% sensitivity, while 83.28% accuracy, 91.15% specificity, and 86.82% sensitivity values were obtained with the training from scratch. According to these results, it is observed that using the learned weights of pre-trained architectures achieves superior performance over the training-from-scratch approach. Accordingly, it was decided to continue using pre-learned weights in the next experimental studies.

We used pre-trained deep architectures as feature extractors in the second experimental work. Accordingly, 1000 deep features were extracted for each deep model using the fully connected layers of the DenseNet201, DarkNet53, and Xception architectures. The deep features of each architecture were given separately to a KNN-based random subspace ensemble classifier input, and the training process was carried out. The confusion matrices obtained from this experimental study are shown in Fig. 5.

According to the confusion matrices given in Fig. 6, the performance metrics obtained from deep architectures are given in Table 4.

As can be seen from Table 4, the pre-trained deep architecture with the highest accuracy score was obtained with the DenseNet201 architecture as 97.07%. In addition, DarkNet53 and Xception architectures produced 94.32% and 95.85% accuracy scores, respectively. On the other hand, the ensemble classifier based DarkNet53 architecture, which is used as a feature extractor, has achieved approximately 4% higher performance than the DarkNet53 architecture based on the fine-tuning approach. These findings showed that a KNN-based

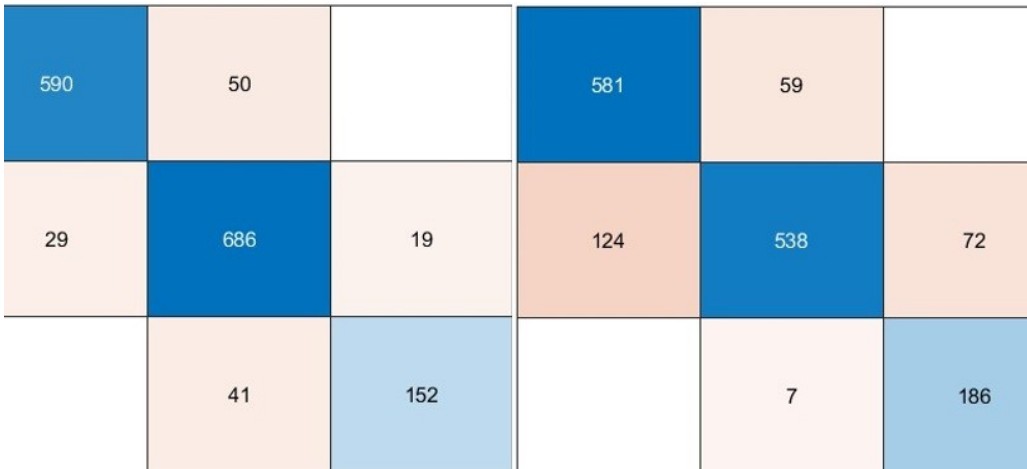

**Figure 5** Performance metrics (%) of hybrid pre-trained deep networks.

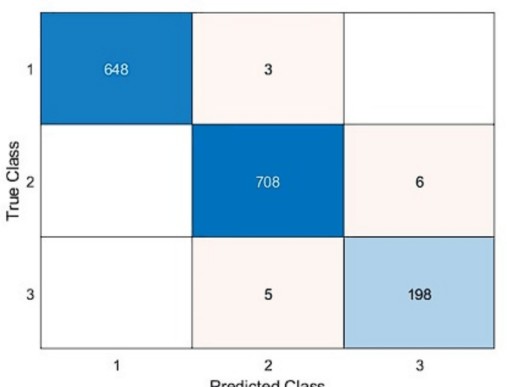 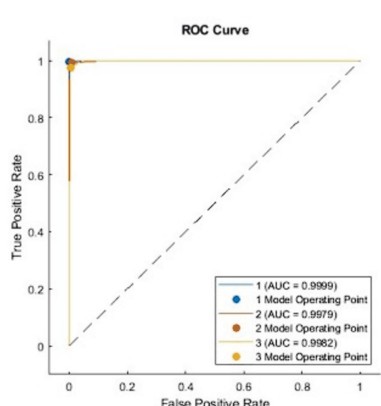

**Figure 6** Confusion matrixes of the proposed model. (Left) Confusion matrix, (right) ROC diagram.

random subspace ensemble classifier model achieved more successful performance than the SoftMax classifier.

In the previous experimental study, the features obtained from the deep architectures used were combined according to different combinations and fed as the input to the classifier model. Therefore, the performance results obtained from this study are given in Table 5.

As can be seen from Table 5, the best model achieved an accuracy score of 99.11% with the combination of DenseNet201, DarkNet53, and Xception architectures. In addition, the second-best model is 98.47%, combining DenseNet201 and DarkNet53 models. These results proved that combining deep features from pre-trained networks with different structures positively affects classification performance. The confusion matrix and roc diagram of the proposed model based on deep hybrid features and a KNN-based random subspace ensemble classifier is illustrated in Fig. 6.

**Table 5 Performance metrics (%) of hybrid pre-trained deep networks.**

| Hybrid models | Accuracy | Sensitivity | Specificity | Precision | F1-Score |
|---|---|---|---|---|---|
| DenseNet201+ DarkNet53 | 9847 | 9756 | 9914 | 9826 | 9790 |
| DarkNet53 + Xception | 9726 | 9689 | 9853 | 9656 | 9672 |
| Xception + DenseNet201 | 9802 | 9678 | 9894 | 9733 | 9705 |
| All models | 9911 | 9875 | 9954 | 9865 | 9870 |

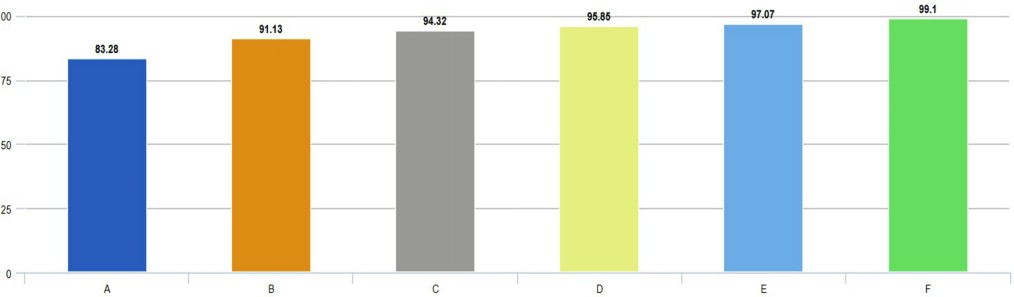

**Figure 7 Comparison of accuracies obtained from all experimental works.**

According to the confusion matrix of the proposed model shown in Fig. 6A, 3, 6, and 5 are misclassified for AD, healthy, and PD classes, respectively. In addition, the AD and healthy classes of the proposed model are correctly classified with 99% and above performance. As a result, the accuracy scores obtained from all experimental studies were compared and given in Fig. 7.

In Fig. 7, accuracy scores from extensive experimental studies are compared. These results show that the proposed model is more successful than other deep network approaches. On the other hand, the fine-tuning approach based on pre-learned weights achieved 8% higher accuracy than the deep neural network approach trained from scratch. Additionally, deep networks based on ensemble classifiers have higher accuracy scores than pre-trained deep models based on the transfer learning approach.

## DISCUSSION

The importance of AI-based diagnosis for Alzheimer's and Parkinson's diseases cannot be overstated. The potential benefits of early detection, accurate diagnosis, personalized treatment, and research advancements are significant and can lead to improved patient outcomes and a better understanding of these diseases. In addition, the development of AI-based diagnosis methods can drive technological advancements in healthcare, leading to the development of new tools and techniques that can be applied to other areas of medicine. For this purpose, many studies based on MRI images have been carried out in the literature. Previous studies based on the dataset utilized in the current study are given in Table 6, and the accuracy results are compared.

**Table 6  Comparison of performance results of previous studies with the proposed model.**

| References | Problem | Accuracy (%) |
|---|---|---|
| *Helaly, Badawy & Haikal (2022)* | Dementia of AD | 97 |
| *Bhagat et al. (2023)* | Dementia of AD | 96.6 |
| *Sivaranjini & Sujatha (2020)* | PD and healthy control | 88.9 |
| *Peng et al. (2017)* | PD and healthy control | 85.8 |
| *Raees & Thomas (2021)* | MCI, AD, and normal | 90.02 |
| *Basnin et al. (2021)* | PD and healthy control | 90 |
| *AlSaeed & Omar (2022)* | AD and healthy control | 99 |
| *Savaş (2022)* | Normal, MCI, and AD | 92.98 |
| *Noella & Priyadarshini (2023a)* | AD, FTD, PD and healthy control | 97.7 |
| *Noella & Priyadarshini (2023b)* | PD, AD, and healthy control | 90.3 |
| *Alsharabi et al. (2023)* | PD, AD, and healthy control | 96.5 |
| Our Model (2023) | PD, AD, and healthy control | 99.10 |

Table 6 presents the accuracy scores from previous studies based on neurology disease diagnosis. Most of these studies have addressed either Alzheimer's or Parkinson's disease. Like the current study, the remaining studies are based on the PD, AD, and healthy control classification, and according to the accuracies obtained from previous studies, *AlSaeed & Omar (2022)* achieved the best performance for AD and healthy control classification with an accuracy score of 99%. In other studies (*Helaly, Badawy & Haikal, 2022*; *Bhagat et al., 2023*; *Noella & Priyadarshini, 2023a*; *Alsharabi et al., 2023*), a performance of 95% and above was achieved, whereas in the remaining studies, accuracy scores between %85-90 were obtained. Based on all these results, the proposed model for classifying PD, AD, and healthy control has produced superior performance over previous studies. As a result, in the current study, a system with early, accurate, low-cost, and objective diagnosis is presented for identifying neurodegenerative diseases such as Alzheimer's and Parkinson's.

In this study, the effect of the resolution enhancement process performed in the preprocessing stage of the proposed system on the classification performance is analyzed in detail. Resolution enhancement has the potential to affect the learning capacity of the model by providing a higher resolution representation of the input data. In this context, various experimental methods were applied to understand the effect of resolution enhancement on classification performance. The proposed system was evaluated with the results obtained without applying VSDR architecture and an accuracy score of 98.40% was obtained. On the other hand, the adoption of the super-resolution-based approach of the proposed system resulted in an increase of about 1% in the accuracy score. This shows that the resolution enhancement process has a positive contribution to the basic performance of the system. In conclusion, these findings highlight the potential benefits of resolution enhancement for classification performance and provide an important basis for future similar studies.

## CONCLUSIONS

This article discussed artificial intelligence-based automatic diagnosis using Alzheimer's disease, Parkinson's disease, and healthy control MRI images. First, MR images were

enhanced and developed using the proposed VSDR network. Then, high-performance low and high-level features were extracted using pre-trained deep models. In the current study, pre-trained deep architectures are used to give higher performance than a newly designed CNN model. This has been proven in experimental studies. Finally, the obtained features were given to the input of the classifier model. A high-performance KNN-based random subspace ensemble classifier model is presented in the classifier model. The proposed architecture achieved high performance in the experimental results compared to the previous studies. Experimental investigations have demonstrated that utilizing the proposed architecture can aid specialist physicians in rapidly, precisely, inexpensively, and objectively categorizing Alzheimer's disease, Parkinson's disease, and healthy, thereby substantially diminishing the required time for diagnosis. Future research will focus on the feature selection and parameter optimization processes based on the proposed model.

### Funding

This study is supported *via* funding from Prince Sattam bin Abdulaziz University project number (PSAU/2023/R/1445). The funders had no role in study design, data collection and analysis, decision to publish, or preparation of the manuscript.

### Grant Disclosures

The following grant information was disclosed by the author:
Prince Sattam bin Abdulaziz University: PSAU/2023/R/1445.

### Competing Interests

The author declares there are no competing interests.

### Author Contributions

- Adi Alhudhaif conceived and designed the experiments, performed the experiments, analyzed the data, performed the computation work, prepared figures and/or tables, authored or reviewed drafts of the article, and approved the final draft.

### Data Availability

This dataset is avaialble at Kaggle: https://www.kaggle.com/datasets/farjanakabirsamanta/alzheimer-diseases-3-class.

### Supplemental Information

Supplemental information for this article can be found online at http://dx.doi.org/10.7717/peerj-cs.1862#supplemental-information.

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
