# Peer review of "A novel approach to recognition of Alzheimer’s and Parkinson’s diseases: random subspace ensemble classifier based on deep hybrid features with a super-resolution image"

_PeerJ Computer Science, doi:10.7717/peerj-cs.1862_

## Round 0.1 · original submission · Major Revisions

The reviewers have substantial concerns about this manuscript. The authors should provide point-to-point responses to address all the concerns and provide a revised manuscript with the revised parts being marked in different color.

**Language Note:** The review process has identified that the English language must be improved. PeerJ can provide language editing services - please contact us at copyediting@peerj.com for pricing (be sure to provide your manuscript number and title). Alternatively, you should make your own arrangements to improve the language quality and provide details in your response letter. – PeerJ Staff

Reviewer 1 ·

Basic reporting

1. While the manuscript is written in English and generally adheres to professional standards, there are instances of awkward phrasing and minor grammatical errors that hinder clarity. For instance, in the introduction, the sentence structure in "These technologies can help with early diagnosis, more accurately classify, and improve patients' access..." is somewhat convoluted and could be rephrased to "These technologies can aid in early diagnosis, enhance classification accuracy, and improve patient access to appropriate treatments."

2. There are minor typographical errors throughout the paper. For instance, "Tablo 1" should be corrected to "Table 1" (line 65). Such errors, while not impeding understanding, detract from the paper's professionalism.

3. While the manuscript references relevant studies, a deeper analysis comparing the proposed method with existing ones would strengthen the background. For example, in the Introduction (lines 74-78), more detailed comparisons with the mentioned studies could provide a clearer picture of how the proposed method advances the field.

Experimental design

1. The manuscript briefly describes the KNN-based random subspace ensemble classifier (lines 196-212), but it lacks depth in explaining the configuration of this classifier. How were the subsets of features chosen for each ensemble member? What was the rationale behind the chosen number of neighbors in KNN? This information is important to assess the robustness of the classification approach.

2. The paper should include a discussion on measures taken to prevent overfitting. Were techniques like dropout, regularization, or early stopping employed in the neural network training process? Overfitting is a common issue in deep learning models, especially when dealing with medical images.

3. The paper mentions employing data augmentation techniques (line 237) but lacks specific details on how these techniques were applied. For example, what were the parameters for rotation, flipping, or brightness adjustment? This information is crucial for replicability and understanding the potential impact on the results.

Validity of the findings

1. The paper presents various performance metrics (lines 224-225), but it does not mention if any statistical tests were conducted to ascertain the significance of the results. For instance, were the differences in accuracy between the proposed method and existing methods statistically significant?

2. The manuscript would benefit from a more detailed description of the validation process. Were cross-validation techniques used? If so, how was this implemented, and what was the rationale behind the chosen approach?

3. The paper uses a publicly available dataset (line 233), but there's no discussion on the representativeness of this dataset. It's essential to know if the dataset adequately represents the diverse demographics and stages of Alzheimer's and Parkinson's diseases. This information is crucial for assessing the generalizability of the findings.

Reviewer 2 ·

Basic reporting

The research paper titled "A novel approach to Recognition of Alzheimer's and Parkinson's Diseases: Random subspace ensemble classifier based on deep hybrid features with a Super-resolution image." This paper discusses the potential of artificial intelligence (AI) technologies in classifying neurodegenerative diseases such as Alzheimer's and Parkinson's. The authors propose a deep hybrid network that combines an ensemble classifier and convolutional neural network to classify Alzheimer's disease, Parkinson's disease, and healthy MRI images. They utilize a deep super-resolution neural network to improve the resolution of MR images and extract low and high-level features using the hybrid deep convolutional neural network. The extracted features are then inputted into a KNN-based random subspace ensemble classifier. Experimental results show that the proposed model achieves high accuracy, sensitivity, specificity, precision, and F1-Score performance values. The author suggested that this AI system has the potential to provide valuable diagnostic assistance in clinical settings. The document also includes reviewing tips and the introduction and abstract of the research paper. The topic of the paper is great, however, I have a few concerns that would like the author to address:

Typo:
1.Line 27 MR images (MRI images?)
2.Line 89 Mr images (MRI & MR image(Line 91)?)
3.Tablo 1 (Table 1)
Questions & Concerns:
1.Please double check the draft and correct the typos and maintain consistency in the use of academic terms (e.g. MRI & MR images?).
1.It is necessary to evaluate the performance of the classifier by k-fold cross validation. Please offer additional information on the results of k-fold cross validation for the classifier.
2.Please provide complete code for performing the classifier, which could repeat the results as mentioned in the paper.

Experimental design

The research paper titled "A novel approach to Recognition of Alzheimer's and Parkinson's Diseases: Random subspace ensemble classifier based on deep hybrid features with a Super-resolution image." This paper discusses the potential of artificial intelligence (AI) technologies in classifying neurodegenerative diseases such as Alzheimer's and Parkinson's. The authors propose a deep hybrid network that combines an ensemble classifier and convolutional neural network to classify Alzheimer's disease, Parkinson's disease, and healthy MRI images. They utilize a deep super-resolution neural network to improve the resolution of MR images and extract low and high-level features using the hybrid deep convolutional neural network. The extracted features are then inputted into a KNN-based random subspace ensemble classifier. Experimental results show that the proposed model achieves high accuracy, sensitivity, specificity, precision, and F1-Score performance values. The author suggested that this AI system has the potential to provide valuable diagnostic assistance in clinical settings. The document also includes reviewing tips and the introduction and abstract of the research paper. The topic of the paper is great, however, I have a few concerns that would like the author to address:

Typo:
1.Line 27 MR images (MRI images?)
2.Line 89 Mr images (MRI & MR image(Line 91)?)
3.Tablo 1 (Table 1)
Questions & Concerns:
1.Please double check the draft and correct the typos and maintain consistency in the use of academic terms (e.g. MRI & MR images?).
1.It is necessary to evaluate the performance of the classifier by k-fold cross validation. Please offer additional information on the results of k-fold cross validation for the classifier.
2.Please provide complete code for performing the classifier, which could repeat the results as mentioned in the paper.

Validity of the findings

The research paper titled "A novel approach to Recognition of Alzheimer's and Parkinson's Diseases: Random subspace ensemble classifier based on deep hybrid features with a Super-resolution image." This paper discusses the potential of artificial intelligence (AI) technologies in classifying neurodegenerative diseases such as Alzheimer's and Parkinson's. The authors propose a deep hybrid network that combines an ensemble classifier and convolutional neural network to classify Alzheimer's disease, Parkinson's disease, and healthy MRI images. They utilize a deep super-resolution neural network to improve the resolution of MR images and extract low and high-level features using the hybrid deep convolutional neural network. The extracted features are then inputted into a KNN-based random subspace ensemble classifier. Experimental results show that the proposed model achieves high accuracy, sensitivity, specificity, precision, and F1-Score performance values. The author suggested that this AI system has the potential to provide valuable diagnostic assistance in clinical settings. The document also includes reviewing tips and the introduction and abstract of the research paper. The topic of the paper is great, however, I have a few concerns that would like the author to address:

Typo:
1.Line 27 MR images (MRI images?)
2.Line 89 Mr images (MRI & MR image(Line 91)?)
3.Tablo 1 (Table 1)
Questions & Concerns:
1.Please double check the draft and correct the typos and maintain consistency in the use of academic terms (e.g. MRI & MR images?).
1.It is necessary to evaluate the performance of the classifier by k-fold cross validation. Please offer additional information on the results of k-fold cross validation for the classifier.
2.Please provide complete code for performing the classifier, which could repeat the results as mentioned in the paper.

Additional comments

The research paper titled "A novel approach to Recognition of Alzheimer's and Parkinson's Diseases: Random subspace ensemble classifier based on deep hybrid features with a Super-resolution image." This paper discusses the potential of artificial intelligence (AI) technologies in classifying neurodegenerative diseases such as Alzheimer's and Parkinson's. The authors propose a deep hybrid network that combines an ensemble classifier and convolutional neural network to classify Alzheimer's disease, Parkinson's disease, and healthy MRI images. They utilize a deep super-resolution neural network to improve the resolution of MR images and extract low and high-level features using the hybrid deep convolutional neural network. The extracted features are then inputted into a KNN-based random subspace ensemble classifier. Experimental results show that the proposed model achieves high accuracy, sensitivity, specificity, precision, and F1-Score performance values. The author suggested that this AI system has the potential to provide valuable diagnostic assistance in clinical settings. The document also includes reviewing tips and the introduction and abstract of the research paper. The topic of the paper is great, however, I have a few concerns that would like the author to address:

Typo:
1.Line 27 MR images (MRI images?)
2.Line 89 Mr images (MRI & MR image(Line 91)?)
3.Tablo 1 (Table 1)
Questions & Concerns:
1.Please double check the draft and correct the typos and maintain consistency in the use of academic terms (e.g. MRI & MR images?).
1.It is necessary to evaluate the performance of the classifier by k-fold cross validation. Please offer additional information on the results of k-fold cross validation for the classifier.
2.Please provide complete code for performing the classifier, which could repeat the results as mentioned in the paper.

Reviewer 3 ·

Basic reporting

In this manuscript, the author introduces an AI-based diagnostic model for Alzheimer's and Parkinson's diseases, combining a deep hybrid network with a super-resolution neural network and a KNN-based random subspace ensemble classifier. This approach achieves high accuracy, sensitivity, and specificity, demonstrating potential for early and accurate disease diagnosis in clinical settings. The study leverages pre-trained deep learning architectures and compares favorably with previous research, underscoring its innovative contribution to neurodegenerative disease detection.

Experimental design

The comparision and benchmarking of the super-resolution image is missing, the author should provide exhaustive analysis on how this deep-learning based image upsampling can help the neural network to better extract the information. A direct and detailed comparison of the image that before and after “increasing” resolution should also be performed for empirical illustration.

Validity of the findings

No comment

Additional comments

In Line 65, it should be “Table 1” rather than “Tablo 1”.

The data showed in table 4 and 5 missed the decimal points.

---

## Round 0.2 · Minor Revisions

Reviewers are generally satisfied with the revisions except for one minor typo that needs to be corrected.

Reviewer 2 ·

Basic reporting

Please check and correct the typo on line 31.

Experimental design

no

Validity of the findings

no

Additional comments

no

Reviewer 3 ·

Basic reporting

The authors addressed my concern and I thus recommend the publication with its current form.

Experimental design

no comment

Validity of the findings

no comment

Additional comments

no comment

---

## Round 0.3 · accepted · Accept

All concerns have been addressed and I suggest accepting this manuscript.